# Antimicrobial Stewardship in College and University Health Settings: A Public Health Opportunity

**DOI:** 10.3390/antibiotics11010089

**Published:** 2022-01-12

**Authors:** Kathryn L. Dambrino, Montgomery Green

**Affiliations:** 1College of Health Sciences and Nursing, Belmont University, Nashville, TN 37212, USA; 2College of Pharmacy, Belmont University, Nashville, TN 37212, USA; montgomery.green@belmont.edu

**Keywords:** antimicrobial stewardship, antibiotic stewardship, antibiotic resistance, college health, university health, college students, university students, outpatient antimicrobial stewardship, public health

## Abstract

Antimicrobial resistance poses a significant threat to public health and safety across the globe. Many factors contribute to antibiotic resistance, most especially are the concerns of excessive prescribing and misuse of antibiotics. Because patient expectations for antibiotics may contribute to prescriber pressures, experts recommend targeting antimicrobial stewardship (AMS) education efforts towards prescribers as well as patients in outpatient settings. Undergraduate university students are a unique and promising target population for AMS efforts because they are in a transformative life stage of social, cognitive, and physical development in which they are learning to independently care for themselves without the presence or influence of parents. By introducing AMS education during this transition, university students may adopt positive antibiotic use behaviors that they will carry throughout their lives. Not only will their personal health be improved, but widespread adoption of AMS in university settings may have a broader effect on public health of present and future generations. Despite public health opportunities, minimal research has examined AMS in university health settings. This article explores current evidence on knowledge, attitudes, and use of antibiotics among university students and discusses opportunities for AMS initiatives in college and university health settings.

## 1. Introduction

Antimicrobial resistance poses a significant threat to public health and safety across the globe. According to the Centers for Disease Control and Prevention (CDC), patients in the United States incur over 2.8 million antibiotic-resistant infections annually, resulting in over 35,000 deaths [1]. In addition, antibiotic-resistant infections remain a financial burden in the U.S. with an estimated $4.6 billion per year in medical care costs [2]. For nearly two years, the SARS-CoV-2 virus has overwhelmed healthcare systems worldwide, leading to direct and indirect consequences for antibiotic resistance [3]. While an increase in healthcare-associated infections was reported in some ICUs during the COVID-19 pandemic, enhanced public awareness of infectious disease may create new opportunities [3,4]. Now, more than ever, medical and public health professionals are called upon to engage in antimicrobial stewardship (AMS) efforts to combat the complex issue of antibiotic resistance in inpatient and outpatient settings.

Many factors contribute to antibiotic resistance, most especially are the concerns of excessive prescribing and misuse of antibiotics. Outpatient clinical settings account for approximately 80–90% of antibiotic prescriptions, and half of these prescriptions are deemed inappropriate or unnecessary [5]. Multidisciplinary AMS strategies have been introduced to improve prescribing practices and combat preventable infections. Electronic surveillance software has contributed to quality data monitoring and a decrease in preventable infections, especially in the inpatient setting [6,7,8]. AMS teams utilizing electronic surveillance software achieve more efficient and accurate data collection, monitoring, and reporting [4]. Other effective AMS strategies include educational and behavioral interventions aimed at prescribers, pharmacists, and nurses [9]. Although these AMS strategies have been recommended in the outpatient setting, challenges to successful implementation remain [1,5,10]. Outpatient clinics may lack the financial resources or information technology (IT) support necessary to implement an electronic surveillance system. Furthermore, clinic visit time constraints and knowledge gaps regarding standard practice guidelines contribute to inappropriate or unnecessary antibiotic prescriptions [11,12,13]. Outpatient medical providers often perceive pressure from patients to prescribe antibiotics and fear poor patient satisfaction rates as well as disruption of the therapeutic provider-patient relationship if antibiotics are not prescribed [9,11,12,13]. Despite widespread documentation of these challenging circumstances, evidence identifying effective solutions is limited.

To address the issue of patient expectations for antibiotics, experts recommend targeting AMS education efforts towards patients as well as prescribers [9,11,12,13,14,15,16]. Because adult patients’ knowledge, attitudes, and beliefs related to antibiotic use are shaped by previous encounters with medical providers, introduction of AMS initiatives may produce better outcomes if started at an early age [9,10,15]. University students, who are typically well-educated young adults aged 18–25, are a unique and promising target population for AMS efforts for a variety of reasons. For many students, college or university is the first time they will seek medical assistance without the guidance of their parents. University students are learning to independently navigate the healthcare system, interact with medical providers, and develop new health behaviors, and their university healthcare experiences may negatively or positively influence their future health practices throughout adulthood [17]. Notwithstanding the public health opportunities, minimal research has explored antibiotic use and AMS in the university student population or on university campuses. The purposes of this article are to (1) explore current evidence on knowledge, attitudes, and usage of antibiotics among university students and (2) discuss opportunities for AMS initiatives in college and university health settings.

## 2. Antibiotic Use and Misuse among University Students

Although the typical university student is young and healthy, many students will seek out medical attention for illness or injury at some point during their studies. In the American College Health Association’s National College Health Assessment survey, university students consistently report respiratory illnesses/infections, urinary tract infections (UTIs), and acne among the most frequent medical diagnoses received from a healthcare provider [18]. Young people aged 15–24 accounted for nearly half of the 26 million sexually transmitted diseases (STDs) diagnosed in the U.S. in 2018, which contributes to the rising prevalence of antibiotic-resistant gonorrhea [1,19]. Because antibiotics are often prescribed to university students for these common chief complaints, it is important to review evidence of appropriate or inappropriate antibiotic use in this population.

University students participating in a qualitative study in United Arab Emirates admitted to engaging in frequent antibiotic use/misuse citing several influencing factors, such as easy access to antibiotics without a prescription, advice from friends/family, time constraints associated with classes, and clinic visit costs [20]. Students may obtain antibiotics legitimately and illegitimately from a variety of sources. While some studies revealed that approximately 16–64% of students self-medicated with antibiotics received from friends and family or leftover from a previous illness [17,21,22], others found this behavior in less than 6% of their student samples with most students having obtained antibiotics from healthcare providers [23,24]. According to Zoorob et al., U.S. university students with the following characteristic were most likely to seek medical care for common cold symptoms: upper class (juniors and seniors), older age, and those using the campus student health clinic [25].

Not only is antibiotic resistance a risk with increased antibiotic use in this population, but also antibiotic-related adverse drug events are a concern. In a 2018 U.S. study, young adults aged 20–34 were twice as likely as older adults to seek medical attention for antibiotic-related adverse drug events at the emergency department [26]. This evidence suggests that students may underestimate the risks associated with consuming antibiotics. Because students often seek medical care at campus student health clinics, it is imperative that evidence-based prescribing practices and education on antibiotic risks be consistently provided. Patients’ antibiotic use behaviors are shaped by previous experiences with medical providers, which makes encounters at university student health clinics especially critical to students’ future use behaviors [9].

### 2.1. Global Context for Antibiotic Use and Misuse

Cultural context must also be considered when examining antibiotic use and misuse of university students from different countries. The presence of infectious illnesses, access to antibiotics, surveillance, and government regulations are significantly different in low- and middle-income countries (LMICs) compared to high-income countries (HICs).

#### 2.1.1. Low- and Middle-Income Countries

According to Tattevin et al., LMICs located in Sub-Saharan Africa and Asia are disproportionately affected by infectious disease and antimicrobial resistance, which is likely a result of weak government regulations, counterfeit and over-the-counter sales of antimicrobials, lack of surveillance/diagnostic testing capability, and poor prescribing practices [27]. In a systematic review and meta-analysis exploring antibiotic use practices of university students in LMICs, Xu et al. concluded that self-medication with antibiotics in these countries is a common occurrence [28]. Of the LMICs included, Africa ranked highest in self-medication practices (55.3%) and South America ranked lowest (38.3%) [28]. Higher education level and previous experience self-medicating with antibiotics were identified as leading risk factors for inappropriate self-medication practices in LMICs [28]. While lack of awareness of antibiotic risks was recognized as a potential cause, increased antibiotic self-medication practices by university students may be a result of widespread availability of antibiotics that can legally be purchased at pharmacies without a prescription [21,22,28]. Government policies and enforcement of regulations are needed to address excessive consumption of antibiotics in countries where antibiotics are available for purchase without a prescription [21,22,28]. Because individuals in LMICs often lack access to appropriate antibiotics when needed for infectious illness, attempts to regulate consumption of inappropriate antibiotic use may be challenging [27].

#### 2.1.2. High-Income Countries

Individuals living in HICs, such as the United States, United Kingdom, and Australia, are estimated to consume the highest number of antibiotics with 25 defined daily doses (DDD) per 1000 persons per day compared to 10–20 DDD per 1000 persons per day in LMICs [27]. Though resources dedicated to AMS in LMICs are minimal to nonexistent, many HICs experience insufficiencies related to human resources for AMS [27]. In HICs, individuals typically require a prescription to obtain antibiotics and surveillance/diagnostic labs are regularly available, yet antimicrobial resistance remains a problem [27]. Though China is mostly considered a middle- to high-income country, Peng et al. highlighted the differences in antibiotic use that may occur in rural or less developed regions of HICs [22]. When comparing university students in the less-developed Guizhou Province to the developed Zhejiang Province, Guizhou students engaged in higher prescribed antibiotic use (79.8% vs. 56.2%) and self-medication practices (33.0% vs. 16.1%) [22]. Overall, university students in both provinces reported antibiotic misuse behaviors with high numbers of over-the-counter antibiotic purchases (Guizhou 73.9% vs. Zhejiang 63.4%) [22]. More evidence exploring the specific cultural challenges facing university students in HICs and LMICs is needed.

## 3. Students’ Knowledge and Attitudes toward Antibiotic Use and Misuse

Evidence directly quantifying the use and misuse of antibiotics in university students is limited; therefore, most studies have focused on students’ self-reported knowledge and attitudes. In a survey-based study exploring common myths of antibiotic use, Shahpawee et al. found 51% of students in Brunei were adequately knowledgeable about antibiotics and antimicrobial resistance [23]. Although many of these students understood that antibiotics are the necessary treatment for UTIs, skin infections, and gonorrhea, they incorrectly believed antibiotics typically treat sore throat, cold/flu, diarrhea, and fever [23]. In a U.S. study, 77% of participating students reported typical symptoms of common cold, such as discolored mucus and low-grade fever, would prompt them to use antibiotics [25]. Several additional studies similarly provided evidence that many college students incorrectly believe antibiotics are the correct treatment for symptoms of viral infections [17,20,21,25,29,30]. Another knowledge deficit was observed in university students’ understanding of antimicrobial resistance as evidence suggested many students mistakenly believed the human body, rather than bacteria, could become resistant to antibiotics [20,23,24]. Though students may misunderstand the concept of antibiotic resistance, a study from the United Kingdom revealed that students recognized antibiotic resistance as a more essential global public health issue than other issues like climate change, food security, obesity, and gender inequality [24]. These findings suggest that increased emphasis on the global and public health effects of antibiotic use may be a worthwhile strategy to motivate this population to engage in behaviors that minimize antibiotic resistance. University students’ expectations for antibiotics may be a factor. Haltwinger et al. reported that U.S. students who received antibiotic prescriptions were more likely to report satisfaction with the student health clinic experience [17]. Since patients who expect antibiotic treatment frequently pressure providers to prescribe antibiotics, student health clinic providers are at risk for engaging in inappropriate prescribing practices that negatively impact students’ future expectations and use of antibiotics [11]. By employing skilled and therapeutic communication techniques to counter these expectations, providers can potentially improve students’ knowledge and attitudes toward appropriate antibiotic use leading to positive antibiotic use behaviors in the future [13].

### Students in Health-Related Disciplines

Knowledge and attitudes of university students in health-related majors (i.e., medicine, nursing, pharmacy, veterinary, public health, and biology) may differ from the general university student population. According to several studies, students in health-related majors consistently demonstrated higher levels of knowledge and attitudes regarding antibiotic use/misuse and antibiotic resistance [21,23,24,30,31]. For example, Dyar et al. determined 92% of a sample of students studying human and animal health in the United Kingdom were aware that most sore throats, coughs, and colds resolve without antibiotic treatment [24]. When comparing antibiotic use behaviors in health-related majors with students not in health-related majors, findings varied. Most studies determined that health-related students scored significantly higher on appropriate antibiotic use behaviors compared to non-health-related students [22,29,30]. However, Pan et al. reported significantly higher rates of inappropriate self-medication with antibiotics among medical students in southern China who had received education on antibiotics and attributed this finding to “a false sense of confidence in self-diagnosis and self-management” [21] (p. e41314). Students in health-related fields may also have a limited understanding of AMS. Although 95% of health-related students in one study deemed inappropriate prescribing, dispensing, and administering of antibiotics to be unethical, authors discovered that only 44% were aware of AMS [24]. These students are preparing to become the future leaders and providers of healthcare across the globe making it an essential public health priority that evidence-based AMS be incorporated into formal training. Table 1 provides a summary of articles studying antibiotic knowledge, attitudes, and use behaviors in college and university students.

## 4. AMS in College and University Settings

### 4.1. Health Literacy and Shared Decision-Making

Though university students are engaged in formal education, they are at risk for low health literacy regarding use of antibiotics and antibiotic resistance. Students have potential during their transitional college years to improve their health literacy related to AMS, resulting in lifelong reduction of expectations for and consumption of unnecessary antibiotics [16]. Shared decision-making is recognized as the preferred approach for educating university students about AMS and empowering them to embrace behaviors that will decrease antibiotic resistance [32]. Shared decision-making involves a collaborative interaction between provider and patient in which the patient takes a more active role in their healthcare plan [32]. Patients are given sufficient evidence and information so they may consider risks and benefits and apply their own values when agreeing to the provider’s recommended plan for their condition [32]. In order for shared decision-making to be effective, the level of AMS health literacy must be considered for both the healthcare provider and the patient [9,15,16]. Therefore, healthcare providers caring for this population should employ practices to enhance health literacy and shared decision-making, such as up-to-date prescribing guidelines, plain language communication, understandable evidence-based written education, “teach-back” techniques, and “watchful waiting” discussions [9,11,15,16,32].

In a pilot quality improvement project, Moes et al. provided an example of using AMS educational tools and shared decision-making to improve knowledge and attitudes for university students in a student health clinic [31]. Students completed a pre-survey prior to their student health clinic visit and completed a post-survey after receiving an educational handout on appropriate antibiotic use along with a verbal review of the information by the healthcare provider [31]. A 10% improvement in knowledge was measured immediately following the educational intervention, and an overall improvement was observed in trust of the healthcare provider and understanding of antibiotic efficacy, risks, and benefits [31]. Since students who did not receive antibiotic prescriptions reported understanding why antibiotics were not warranted, the results highlight the importance (and power) of effective communication between patient and provider [31]. Though a small sample was used in this project, the outcomes support similar finding by Haltiwanger et al., in which students’ satisfaction was significantly more likely when the provider gave a specific diagnosis (*p* < 0.01) and clearly explained the rationale for prescribing or not prescribing an antibiotic (*p* < 0.01) [17].

### 4.2. Opportunities for AMS in College and University Health Settings

AMS is essential to improve antibiotic use among university students, however, studies related to AMS initiatives in college and university health settings are rarely published. Throughout the literature, recommendations for university-based AMS initiatives include provider-focused education to improve prescribing and communication, student-focused education, student health clinic educational handouts, interprofessional learning experiences, and social media campaigns [17,20,21,22,23,24,25,29,30,31,32].

Colleges and universities are optimal settings to apply the CDC’s Core Elements of Outpatient Antibiotic Stewardship, a framework designed for outpatient clinics to help improve antibiotic prescribing and use [11]. The CDC’s Core Elements of Outpatient Antibiotic Stewardship include (1) commitment, (2) action for policy and practice, (3) tracking and reporting, and (4) education and expertise [11]. A model for applying the CDC’s Core Elements of Outpatient Stewardship in the college and university health setting is provided in Figure 1 [11].

Many student health clinics have on-campus pharmacies to facilitate stronger AMS collaboration between pharmacists and prescribers. University clinics and pharmacies located in countries where over-the-counter antibiotics are legally permitted can commit to only dispensing antibiotics to students who have a valid prescription. Universities can also implement a collaborative, multidisciplinary approach to AMS by engaging faculty experts and students in health sciences departments (i.e., biology, public health, nursing, pharmacy, and medicine) in AMS research, program development, and outreach. In addition, student health clinics should incorporate procedures and electronic medical record features to aid in auditing and reporting antibiotic prescribing practices for high-priority diagnoses, such as UTIs and respiratory illnesses [11,15]. Electronic medical record technology can provide real-time feedback to identify problem areas and help providers adhere to prescribing guidelines [9,11,15,16].

Another advantage in student health clinics is accessibility. Typically, student health clinics are conveniently located on campus and visits are of minimal to no cost to students. Providers may experience less pressure and more confidence introducing “watchful waiting” in this setting because students can avoid the added costs and inconvenience associated with typical follow-up appointments in outpatient settings [12]. Because students often seek care from student health clinics, providers can build trusting relationships with students making it easier to effectively communicate why antibiotics are not warranted [13]. A study by Ellis et al. posits that diverse social networks, which are typically found on college campuses, enhance positive self-care health behaviors and likely decrease antibiotic use [33]. Colleges may boost student engagement in AMS efforts by utilizing group and dyadic peer educators among students of diverse backgrounds [15,34].

#### Global Context for AMS in College and University Settings

Global context must be considered before implementing AMS initiatives in the college and university setting. According to Tattevin et al., AMS initiatives that work well in HICs, such as those mentioned in Figure 1, do not always apply well to LMICs [27]. For example, robust surveillance and diagnostic technology may not be available to university health centers in LMICs. Therefore, AMS efforts in LMICs should focus on realistic and achievable outcomes. Tattevin et al. recommended focusing on cost-effective solutions, such as educating and training providers in LMIC on evidence-based antibiotic use and disseminating patient educational materials via the internet [27]. Universities aiming to enhance AMS must create health service infrastructures to enhance utilization of services, especially in LMICs where alternative healthcare access may be scarce or inadequate quality. In a Nigerian study by Abiodun et al., students reported overall positive opinions of the university health services available, and they were more likely to utilize university health services based on staff experience/availability, organization of healthcare, waiting time, continuity of care [35]. As evidenced in a study by Xu et al., higher education levels among students in LMICs may be associated with higher self-medication practices with antibiotics, which indicates a potential deficit in health literacy [28]. Efforts to educate and improve health literacy related to antibiotic use and misuse is imperative in this population. University students in LMICs typically receive higher levels of education than the general population, therefore they often emerge as leaders [28]. Students’ antibiotic use behaviors could significantly influence others in their communities and impact the public health of future generations, which makes university students a high-priority target group for AMS efforts in these countries [28].

## 5. Conclusions

In this article, we provide an overview of current evidence that suggests university students have knowledge and attitude deficits related to safe and appropriate antibiotic use, and they are at risk to develop antibiotics misuse behaviors. We also highlight opportunities for AMS in college and university health settings and include a model for how to apply the CDC’s Core Elements of Outpatient Antimicrobial Stewardship to these settings. Whereas current evidence supports the need for enhanced AMS initiatives in this population, gaps and limitations exist. Additional and more rigorous studies are needed to directly measure antibiotic use and identify specific approaches to AMS in this population. Cultural context and government policies must be considered when studying antibiotic use and misuse in the university student population as the expectation for and availability of antibiotics varies from country to country. We hope this article generates interest in university students as a focus for new outpatient AMS research and practice initiatives.

With 20 million students enrolled in universities across the U.S., the introduction of AMS efforts through on campus student health clinics could have a significant impact on public health [36]. University students are in a transformative life stage of social, cognitive, and physical development in which they are learning to independently care for themselves without the presence or influence of parents [37]. By introducing AMS during this unique life stage, university students may adopt positive antibiotic use behaviors that they will carry throughout their lives. Not only will their personal health be improved, but widespread adoption of AMS in college and university students across the globe could have a broader effect on public health of present and future generations.

## Figures and Tables

**Figure 1 antibiotics-11-00089-f001:**
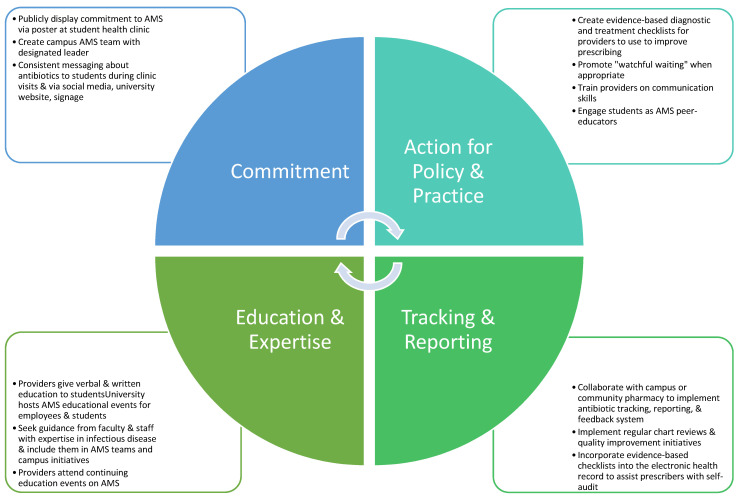
Application of the CDC’s Core Elements of Outpatient Antibiotic Stewardship to the college and university health setting.

**Table 1 antibiotics-11-00089-t001:** Articles on antibiotic knowledge, attitudes, and use behaviors among college and university students.

Author (Year)	Country	Sample	Methods	Findings	Recommendations
Al-Kubaisi et al. (2018) [20]	United Arab Emirates	*n* = 15 college students enrolled in their first year at a healthcare university	Semi-structured face-to-face interviewing to gain an enhance understanding of knowledge, attitude, belief, and experience of college students related to antibiotic use without prescription; Andersen model was applied as the theoretical thematic analysis used to identify, analyze, and report repeated themes within responses	4 themes were identified: (1) Medication habits and practices, (2) reasons for self-medication, (3) access to antibiotics without a prescription, (4) perceptions of antibiotic and the development of resistance; Students had misconceptions that antibiotics are appropriate to use for viral illnesses, and many students reported access to antibiotics without a prescription. Most students were familiar with antibiotic resistance and self-medication of antibiotics worsens resistance. Previous experience with antibiotics was an influencing factor as well as time saving factor, perceived urgency, costs, family/friend advice	Need more research to determine how policy change could improve self-prescribing behaviors, to fully understand student experiences with medication, and to examine legal prohibition of selling antibiotics without prescription; Recommend awareness campaign target towards physician over-prescribing
Blyer et al. (2016) [32]	United States	12 peer-reviewed articles	PRISMA framework used for review and the following inclusion criteria applied: (1) articles about shared decision-making for respiratory infections, (2) articles about college students or adults, (3) articles about antibiotic use for respiratory tract infections	Shared decision-making was preferred among college students and young, educated adults; Some studies suggested shared decision-making was an effective tool for decreasing antibiotic use for respiratory infections	Shared decision-making is a promising strategy for decreasing antibiotic misuse in college students; College health centers should enhance prescribing practices and education to promote a better understanding of AMS in college health; Recommend future research on shared decision-making in college students and international students
Dyar et al. (2018) [24]	United Kingdom	*n* = 255 human and animal health students from 25 universities during the 2016 Antibiotic Guardian campaign	25 question, cross-sectional survey, which assessed knowledge, attitudes, and practices with antibiotic use and awareness of AMS, emailed to students enrolled in different human and animal health courses; Survey also temporarily available on the Public Health England website	Only 5.8% students used antibiotics obtained from friends/family, online source, or leftover from previous illness; 100% knew bacteria could develop resistance to antibiotics and 41% believed human body could develop resistance; 92% were are that most respiratory illnesses improve without antibiotics; Students believed antibiotic resistance was a bigger global challenge (mean of 9 on a scale of 1–10) compared to climate change, food security, gender inequality, and obesity (*p* < 0.001); 44% students were aware of AMS with higher level students more likely to be aware (*p* < 0.01). 20% students believed their knowledge was sufficient for future clinical practice with many requesting more information	Overall, students from diverse healthcare courses were knowledgeable with good attitudes toward antibiotic resistance; Recommend development of curricula that addresses core principles of AMS; Recommend global and national campaigns to increase awareness as well as interprofessional learning experiences to improve AMS
Haltiwanger et al. (2001) [17]	United States	*n* = 129 college students with respiratory symptoms seen in student health clinic in Virginia	30 question survey divided into 2 parts—Part 1 administered pre-visit to assess understanding of illness and knowledge of antibiotics; Part 2 administered post-visit to assess demographics, past antibiotic use and medical visits	71 (55%) students expected an antibiotic prescription; Satisfaction most likely if students received antibiotic (*p* = 0.01) and when specific diagnosis was given (*p* < 0.01) and when students received clear explanation of why antibiotics were or were not necessary (*p* < 0.01)	Appropriate communication is necessary to instill good health behaviors related to antibiotic use; Providers should use alternative terminology that is less suggestive of infection and provide education to students; Educational handout created for college students
Jairoun et al. (2019) [29]	United Arab Emirates	*n* = 1200 college students from Ajman University 600 medical students (case) 600 non-medical students (control)	33 question survey assessed antibiotic knowledge, attitudes, and self-medications practices among students; mixed qualitative and quantitative questions; scores were compared between medical and non-medical students	Students scored highest in attitudes at 76% then knowledge at 59% and practice at 45%; Overall, medical students scored significantly higher than non-medical students in knowledge, attitudes, and practices, respectively (*p* = 0.0001; *p* = 0.000; *p* = 0.002). All students reported overuse of antibiotics for respiratory tract infections	Recommend educational campaigns to address lack of awareness that antibiotic resistance is a national and international problem. Specifically, better education is needed to address knowledge deficits related to antibiotic use for viral respiratory illnesses
Moes et al. (2018) [31]	United States	*n* = 44 college students who visited a college health clinic in Nebraska	19-item survey assessed student knowledge of respiratory infections, especially how antibiotics work against bacteria, viruses, and all germs; students completed the pre-visit survey then received education by the provider with a handout then completed a post-visit survey	Knowledge of antibiotics improved following the educational intervention (*p* = 0.1). Changes in knowledge about efficacy and provider trust was mixed. Knowledge regarding correct use was high before and after. 98% students were satisfied with the visit regardless of antibiotic prescription	Recommend future research involving larger, more diverse samples of college students to determine how educational interventions will improve their knowledge of antibiotics; Recommend analyzing effect of educational intervention at a later interval to ensure knowledge was truly improved and retained. Recommend educational handouts and verbal review by provider about viral respiratory infections and antibiotics
Pan et al. (2012) [21]	China	*n* = 1300 college students from Shantou University	36-question, quantitative and qualitative survey assessed students’ demographics and self-medication behaviors and knowledge of antibiotics; Respondents were divided into 2 groups: (1) PKA group = students with prior knowledge of antibiotics (all medical students except 1st years), and (2) non-PKA group = students without prior knowledge of antibiotics (1st year medical students and non-medical students)	47.8% reported antibiotic self-treatment; Risk factors for self-medication of antibiotics included PKA, older age, and higher monthly allowance; Students commonly self-treated respiratory symptoms and fever with antibiotics; PKA group had better knowledge regarding correct antibiotic use and common adverse reactions (*p* < 0.05); Higher knowledge yet higher rates of self-treatment with antibiotics among the PKA group suggested that PKA may create “a false sense of confidence in self-diagnosis and self-management”	Education was shown to improve the PKA group’s knowledge of appropriate antibiotic use, which suggests education may benefit all students; Recommend targeted education to college students through workshops, seminars, and social media; Recommend stricter government laws to regulate the sale of non-prescribed medications in Chinese pharmacies
Peng et al. (2018) [22]	China	*n* = 3995 college students from developed (Zhejiang) and less developed (Guizhou) regions	Cross-sectional survey assessed antibiotic use behaviors and socio-demographic factors; associations between socio-demographic factors and behaviors were examined	Guizhou students were significantly more associated with misuse of antibiotics, which included antibiotic self-medication (*p* < 0.001), over the counter use (*p* < 0.001), asking for prescription from doctor (*p* < 0.001), and prophylactic use (*p* < 0.001). Students with medical backgrounds were significantly associated with better antibiotic use behaviors, however students with medical parents engaged in poorer antibiotic use behaviors. Higher education level was associated with increased antibiotic misuse	Recommend health education programs on antibiotic use targeted to the general public and college students; Recommend initiatives to eliminate the problem of people taking leftover antibiotics: (1) regulate physician overprescribing, (2) educate patients how to correctly take antibiotics as prescribed, (3) ban antibiotic distribution in pharmacies without prescriptions, (4) instruct patients to throw away leftover antibiotics; Recommend future research to determine causal relationships
Sakr et al. (2020) [30]	Lebanon	*n* = 750 college students 477 health-related majors 273 non-health related majors	Cross-sectional survey assessed knowledge, attitudes, and practices related to antibiotic use and resistance; scores were compared between health students and non-health students	80.2% health students had knowledge related to antibiotics effectiveness to treat bacterial/viral infections compared to 36.9% of non-health students (*p* < 0.001); 94% of health students answered correctly that it is not okay to use friends’ or family members’ antibiotics compared to 85.2% of non-health students (*p* < 0.001); 84.4% health students reported checking expiration of antibiotics prior to use compared to 74.5% non-health students (*p* < 0.001)	Recommend education programs, such as seminars, workshops, and course curricula, targeted toward college students without health background; Encouraged campaigns to promote vaccination and hygiene, public health courses in curricula, media campaigns, and proactive pharmacist roles
Shahpawee et al. (2020) [23]	Brunei Darussalam	*n* = 130 college students at the Universiti Brunei Darussalam 85% from science or health majors	Cross-sectional survey assessed antibiotic use and knowledge	69% had previously used antibiotics with most acquired from healthcare provider. Less than 4% obtained antibiotics from friend/family or from leftover supply; 51% students had good level of knowledge of antibiotic use and resistance (mean score 9 out of 14). 41% had misconceptions that antibiotics were appropriate to treat viral illnesses; 76% believed the body could become resistant to antibiotics	Overall, good level of knowledge was found. Recommend increasing awareness of correct antibiotic use to correct misconceptions. More research is needed in this population
Xu et al. (2019) [28]	Several low- and middle-income countries in Africa (10), Asia (36), Europe (1), and South America (2)	49 peer-reviewed articles	Meta-analysis and systematic review performed; Agency for Healthcare Research and Quality 11-item checklist used to appraise quality	Total prevalence of self-medication with antibiotics (SMA) was 49%; Africa had highest SMA with 55.3% and South America had lowest SMA with 38.3%; More educated students were more likely to engage in SMA	Recommend pharmacological education, national policy and law changes, and improved provider practices in university clinics and hospitals
Zoorob et al. (2001) [25]	United States	*n* = 425 college students from 3 college campuses in Louisiana and Indiana	Cross-sectional surveys asked participants to provide answers regarding their perceptions and use of antibiotics in three different clinical scenarios in which symptoms of different illnesses were described	Use of antibiotics were higher in students who believed antibiotics were the correct treatment for the common cold; Older students were associated with increased antibiotic use (*p* = 0.001) and students utilizing the campus health clinic were more likely to seek care (*p* < 0.001)	Recommend targeted education to providers and students to (1) decrease use of campus health clinic for self-limiting viral illnesses, (2) decrease antibiotic misuse, (3) promote evidence-based options for non-prescription treatments

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
