# Peer review of "Antimicrobial Stewardship in College and University Health Settings: A Public Health Opportunity"

_antibiotics, 2022, doi:10.3390/antibiotics11010089_

Round 1

Reviewer 1 Report

Thank you for this opportunity to revise this manuscript. The topic is interesting and timely. Antimicrobial resistance is recognized as a global public health threat and consequently a systematic approach to improve antimicrobial use like the Antimicrobial stewardship (AMS) is an important field of research to study. Here some suggestions to improve this research.

The introduction section should be expanded. The role of surveillance in providing quality data that can be used in an effective monitoring to reduce the incidence of preventable infections is not mentioned but should be. Some examples worth including are the following:

  • Russo PL, et al. Impact of electronic healthcare-associated infection surveillance software on infection prevention resources: a systematic review of the literature. J Hosp Infect. 2018 May;99(1):1-7. doi: 10.1016/j.jhin.2017.09.002. Epub 2017 Sep 8. PMID: 28893614.
  • Migliara G, et al. Multimodal surveillance of healthcare associated infections in an intensive care unit of a large teaching hospital. Ann Ig. 2019 Sep-Oct;31(5):399-413. doi: 10.7416/ai.2019.2302. PMID: 31304521
  • Mitchell BG, et al. Preventing healthcare-associated infections: the role of surveillance. Nurs Stand. 2015 Feb 10;29(23):52-8. doi: 10.7748/ns.29.23.52.e9609. PMID: 25649603.
  • Behnke M, et al. Information technology aspects of large-scale implementation of automated surveillance of healthcare-associated infections. Clin Microbiol Infect. 2021 Jul;27 Suppl 1:S29-S39. doi: 10.1016/j.cmi.2021.02.027. PMID: 34217465.

In this pandemic situation, it is appropriate discuss its direct and indirect consequences. Specifically, the burden of illnesses caused by antimicrobial-resistant organisms depends on the number and nature of infections, and the availability, effectiveness, and safety of alternative treatments. SARS CoV-2 can influence all these components. Good references are:

  • Baccolini V, et al. The impact of the COVID-19 pandemic on healthcare-associated infections in intensive care unit patients: a retrospective cohort study. Antimicrob Resist Infect Control. 2021 Jun 4;10(1):87. doi: 10.1186/s13756-021-00959-y.
  • Murray, A. K. (2020). The novel coronavirus COVID-19 outbreak: global implications for antimicrobial resistance. Frontiers in microbiology, 11, 1020.

AMS in College and University Settings

I think that it would be a good idea to improve the readability of the manuscript divide the analyzed studies by continents. The cultural aspect influences this field of research and I think that is correct to underline it. Here good references for this aspect:

  • Mostafa, A et al. Is health literacy associated with antibiotic use, knowledge and awareness of antimicrobial resistance among non-medical university students in Egypt? A cross-sectional study. BMJ open, 11(3), e046453
  • Castro-Sánchez, E, et al. Health literacy and infectious diseases: why does it matter? International Journal of Infectious Diseases, 43, 103-110

It could be useful to assess the quality of the studies examined and discuss this aspect.

Author Response

Dear Reviewer,

Thank you for taking reviewing our manuscript titled “Antimicrobial Stewardship in College and University Health Settings: A Public Health Opportunity.” We truly appreciate your thoughtful and helpful inquiries. We have made adjustments in response to your recommendations. All changes in the manuscript are highlighted in yellow (note: you may find changes highlighted that were made in response to a separate reviewer).

Suggestion 1: We expanded the introduction to include the role of surveillance to provide quality data to be used to effectively monitor and reduce preventable infections. We used 3 of the 4 sources provided by the reviewer. These changes are reflected in lines 44-53.

Suggestion 2: We added information on how the COVID-19 pandemic may impact antibiotic resistance, and we incorporated the 2 sources recommended by the reviewer. These changes are reflected in lines 33-38.

Suggestion 3: We expanded the section on Global Context for Antibiotic Use and Misuse and moved it earlier to section 2. It is now labeled as section 2.1. Though we did not divide this section into continents, we believed the readability was enhanced by creating subsections of low- and middle-income countries (LMIC) and high-income countries (HIC). We identified the countries that fell within these categories. These changes are reflected in lines 110-152. We also added the subsection 4.2.1. Global Context for AMS in College and University Settings to provide additional context for how AMS might differ in countries with less resources. These changes are reflected in lines 285-308.

Thank you again for reviewing this manuscript. We appreciate your contribution to enhancing the strength and potential impact of the manuscript.

Reviewer 2 Report

I am delighted to review this manuscript, covering an important aspect of the subject, with a great presentation of results, making this article interested to the readers of the journal Antibiotics. The manuscript follows the scope of the journal Antibiotics.

 I would recommend the article could be published in Antibiotics in the present form.

However, the authors need to address the below-mentioned queries.

 (a)  All the studies are done before 2018,  the author could provide if any current data is available.

(b) The authors could have discussed more global scenarios of stewardship based on the economy and facility of college and university health settings. 

(c)  The author could have used a graph to present the data to provide more visual representation.

Author Response

Dear Reviewer,

Thank you for taking reviewing our manuscript titled “Antimicrobial Stewardship in College and University Health Settings: A Public Health Opportunity.” We truly appreciate your thoughtful and helpful inquiries. We would like to address each of your queries and detail what adjustments we have made in response to your recommendations. All changes in the manuscript are highlighted in yellow (note: you may find changes highlighted that were made in response to a separate reviewer).

Query A: The reviewer queried why all studies were done before 2018. Unfortunately, there are limited studies that examine antibiotic use and antimicrobial stewardship in this population and setting, which is one of the reasons we believe this manuscript is important. For many experts in this field, focus has shifted to the COVID-19 pandemic, which is one of the reasons we suspect there are limited articles published from 2020 onward. We would like to point out that 4 of the articles focusing on university students were published after 2018: Shahpawee et al. (2020), Jairoun et al. (2019), and Sakr et al. (2020), and Xu et al. (2019)

Query B: The reviewer recommended discussion of more global scenarios of stewardship based on the economy and facility of college and university health settings. We agree with this recommendation and decided to expanded the section on Global Context for Antibiotic Use and Misuse and moved it earlier to section 2. It is now labeled as section 2.1. We believe the readability was enhanced by creating subsections of low- and middle-income countries (LMIC) and high-income countries (HIC). We identified the countries that fell within these categories. These changes are reflected in lines 110-152. We also added the subsection 4.2.1. Global Context for AMS in College and University Settings to provide additional context for how AMS might differ in countries with less resources. These changes are reflected in lines 285-308.

Query C: The reviewer recommended more visual representation of the data. We added a table to provide an overview of each article that examined antibiotic use and perceptions among university students. It is labeled Table 1 starting at line 206.

Thank you again for reviewing this manuscript. We appreciate your contribution to enhancing the strength and potential impact of the manuscript.